# Comparative Transcriptome Analysis Reveals the Effect of Lignin on Storage Roots Formation in Two Sweetpotato (*Ipomoea batatas* (L.) Lam.) Cultivars

**DOI:** 10.3390/genes14061263

**Published:** 2023-06-14

**Authors:** Taifeng Du, Zhen Qin, Yuanyuan Zhou, Lei Zhang, Qingmei Wang, Zongyun Li, Fuyun Hou

**Affiliations:** 1Crop Research Institute, Shandong Academy of Agricultural Sciences/Scientific Observing and Experimental Station of Tuber and Root Crops in Huang-Huai-Hai Region, Ministry of Agriculture and Rural Affairs, Jinan 250100, China; a986947745@126.com (T.D.); qinzhenbio@163.com (Z.Q.);; 2Key Laboratory of Phylogeny and Comparative Genomics of the Jiangsu Province, School of Life Sciences, Jiangsu Normal University, Xuzhou 221116, China

**Keywords:** sweetpotato, storage root, transcriptome, lignin synthesis, transcription factors

## Abstract

Sweet potato (*Ipomoea batatas* (L.) Lam.) is one of the most important crops with high storage roots yield. The formation and expansion rate of storage root (SR) plays a crucial role in the production of sweet potato. Lignin affects the SR formation; however, the molecular mechanisms of lignin in SR development have been lacking. To reveal the problem, we performed transcriptome sequencing of SR harvested at 32, 46, and 67 days after planting (DAP) to analyze two sweet potato lines, Jishu25 and Jishu29, in which SR expansion of Jishu29 was early and had a higher yield. A total of 52,137 transcripts and 21,148 unigenes were obtained after corrected with Hiseq2500 sequencing. Through the comparative analysis, 9577 unigenes were found to be differently expressed in the different stages in two cultivars. In addition, phenotypic analysis of two cultivars, combined with analysis of GO, KEGG, and WGCNA showed the regulation of lignin synthesis and related transcription factors play a crucial role in the early expansion of SR. The four key genes *swbp1*, *swpa7*, *IbERF061,* and *IbERF109* were proved as potential candidates for regulating lignin synthesis and SR expansion in sweet potato. The data from this study provides new insights into the molecular mechanisms underlying the impact of lignin synthesis on the formation and expansion of SR in sweet potatoes and proposes several candidate genes that may affect sweet potato yield.

## 1. Introduction

Sweet potato is one of the seventh most important food crops in the world [1] and produces approximately 88.9 million tons of storage root (SR) from an area of 7.4 million ha [2]. China is the largest sweet potato-producing country with 47.8 million tons, which accounts for 53.8% of the total production of sweet potato worldwide. SR is the main edible tissue, its formation and development is the most important agronomic trait in sweet potato producing, which is accompanied by complex biological processes such as adventitious root morphogenesis and accumulation of carbohydrates, storage proteins, and secondary metabolites [3]. SR yield varied in sweet potato cultivars and is prone to environmental change [4,5,6]. It was reported that SR yield is not only dependent upon the rate and duration of SR expansion but also on the beginning of SR formation, the leaves’ longevity, and the growth stage [7,8].

Lignin, a phenylpropanoid compound, plays an important role in the formation of secondary cell walls and is, therefore, considered to play an important role in the formation of sweet potato SRs [9,10,11]. Lignin polymer generally comprises p-hydroxyphenyl (H), guaiacyl (G), and syringyl (S) units in plants, which be synthesized by phenylalanine by a series of enzymes, such as cinnamate 4-hydroxylase (C4H), 4-coumarate CoA ligase (4CL), 4-hydroxycinnamoyl-CoA shikimate/quinate4-hydroxycinnamoyl transferase (HCT), caffeoyl-CoA O-methyltransferase (CCoAOMT), cinnamoyl-CoA reductase (CCR), ferulate 5-hydroxylase (F5H), caffeic acid 3-O-methyltransferase (COMT), cinnamylalcohol dehydrogenase (CAD), and peroxidases (PER). The lignin biosynthetic genes are transcriptionally regulated by the transcriptional regulators. Transcription factors (TF) in the regulation of lignin biosynthesis genes also include NAC [12], MYB [13], and WRKY [14]. Recent studies revealed that miRNA is involved in the regulation of lignin biosynthesis [15,16]. Jin found that calcium-dependent protein kinase (CDPK) mediated methionine adenosyl transferase via the 26S proteasome pathway and affects ethylene biosynthesis and lignin deposition in Arabidopsis [17]. Gibberellin (GA) and cytokinin are reported to be involved in regulating lignin biosynthesis [18,19].

Studies have shown that lignin biosynthesis is connected with root formation and development in sweet potatoes. Firon compared the expression profiles of initiating SRs and fibrous roots and identified the lignin biosynthesis down-regulated at an early stage of SR formation [9]. Ectopic expression of maize Lc regulatory gene in sweet potato induced the expression of lignin biosynthesis genes and affected SR development [11]. Kim discovered many differently expressed genes related to phenylpropanoid biosynthesis in adventitious root formation through RNA-seq analysis [20]. However, the molecular mechanism of lignin in SR development keeps it obscure.

In this study, we selected SR at three developmental stages of two sweet potato varieties, compared the full-length transcriptome data and investigated the gene expression profiles by using full-length and second-generation transcriptome. The primary objective is to reveal the molecular mechanism by which lignin synthesis affects the formation and expansion of sweet potato SR.

## 2. Materials and Methods

### 2.1. Plant Materials

Sweet potato cultivars, “Jishu25 (J25)” and “Jishu29 (J29)”, were planted in the experimental station of the Crops Research Institute, Shandong Academy of Agricultural Sciences, Jinan, China. SRs at the three stages were collected from sweet potato plants 32 days after planting (DAP) (D1), 46 DAP (D2), and 67 DAP (D3) [21], respectively. Three independent biological replicates were taken from each stage of each variety, and each biological replicate came from three independent sweet potato SRs. The samples were immediately frozen in liquid nitrogen and stored at −80 °C for hormone and RNA isolation. At the root developmental stages, the numbers and yield of SR were counted and weighed. The root/shoot (R/S) ratio was calculated according to the root weight divided by the shoot biomass of per plant.

### 2.2. Determination of Lignin Content

Analysis of the lignin content was performed with slight modification according to the methods described by [22]. In short, 10 mg dried roots (W) were digested in 2.5 mL HoAc solution containing 25% (*v*/*v*) acetylbromide and 0.1 mL 70% perchloric acid. The sealed vessel was mixed fully and incubated for 40 min at 80 °C while shaking. After incubation and cooling, the slurry was centrifuged at 23,477× *g* for 15 min. The supernatant was added to 2.5 mL of 2 M NaOH and 1 mL acetic acid. After 20 min, the absorbance (A) was measured at 280 nm. The lignin content was calculated using the following formula: lignin (mg/g) = 0.147 × (ΔA − 0.0068) ÷ W × 50.

### 2.3. RNA Extraction, Full-Length cDNA Library Construction and Sequencing

Total RNAs were extracted using RNA prep Pure Plant plus Kit (Tiangen Biotech (Beijing) Co., Ltd., Beijing, China) and purified with the RNA easy Plant Mini Kit (Qiagen, Valencia, CA, USA). RNA quality was verified using a 2100 Bioanalyzer RNA Nanochip (Agilent, Santa Clara, CA, USA), and quantified using NanoDrop ND-1000 Spectrophotometer (Nano-Drop, Wilmington, DE, USA).

For full-length cDNA library, cDNA was synthesized using SMARTer PCR cDNA Synthesis Kit, and optimized for PCR amplification. The fragments for large-scale PCR were performed using magnetic beads to obtain sufficient total cDNA. The complete SMRT bell library was constructed with using a SMARTer PCR cDNA Synthesis Kit and assembly was performed on the PacBio Sequel platform, the second-generation sequencing and assembly was implemented on the Hiseq 2500 sequencing platform (Illumina) with PE150 by Novogene Co., Ltd. (Beijing, China).

### 2.4. Functional Annotation, Identification and Analysis of DEGs

The full cDNA sequence was processed with SMRTlink7.0 software (https://www.pacb.com/support/software-downloads (accessed on 5 July 2021)) and corrected with the second-generation transcription data to obtain consistent unigenes, then removed any redundancy unigenes by CD-HIT software (4.8.1) (https://github.com/weizhongli/cdhit (accessed on 5 July 2021)) [23]. Unigenes were functionally annotated using the BLASTX alignment (E-value ≤ 10^−5^) against seven databases including GO (Gene Ontology; http://geneontology.org/ (accessed on 5 July 2021)), KEGG (Kyoto Encyclopedia of Genes and Genomes; https://www.kegg.jp/ (accessed on 5 July 2021)), KOG (Eukaryotic Ortholog Groups; ftp://ftp.ncbi.nih.gov/pub/COG/KOG/kyva (accessed on 5 July 2021)), Nr (NCBI non-redundant proteins; https://www.ncbi.nlm.nih.gov/guide/proteins/ (accessed on 5 July 2021)), Nt (NCBI nucleotide sequences; https://www.ncbi.nlm.nih.gov//guide/dna-rna/ (accessed on 5 July 2021)), Pfam (Protein Family Database; http://pfam.xfam.org/ (accessed on 5 July 2021)), and Swiss-prot (a manually annotated and reviewed protein sequence database; https://www.uniprot.org/ (accessed on 5 July 2021)) databases. The best alignment results were selected for the annotation of the unigenes.

Gene expression levels were analyzed by RSEM software (v1.3.3) (http://deweylab.github.io/RSEM/) (accessed on 9 June 2023) [24]. Differential expression genes (DEGs) among SRs of the different developmental stage were identified by DESeq package (1.10.1) with |log2 (FoldChange)| > 0 and q value < 0.05 (http://www.bioconductor.org/packages/release/bioc/html/DESeq.html (accessed on 8 July 2021)) [25]. GO enrichment analysis of DEGs was performed using Goseqsoftberry with corrected and *p* value < 0.05 (http://www.bioconductor.org/packages/release/bioc/html/goseq.html (accessed on 8 July 2021)). KEGG enrichment was analyzed by KOBAS3.0 softberry (http://kobas.cbi.pku.edu.cn/download.php (accessed on 8 July 2021)) with corrected *p* value < 0.05. To further analyze the regulatory mechanism of root formation and development, weighted gene co-expression networks analysis (WGCNA) was performed with the WGCNA package (https://horvath.genetics.ucla.edu/html/CoexpressionNetwork/Rpackages/WGCNA/ (accessed on 25 July 2021)) in the R software (4.3.0) [26], and network visualization for each module was analyzed using the Cytoscape software 3.6.1 (https://cytoscape.org/ (accessed on 26 July 2021)) [27].

### 2.5. Real-Time Quantitative PCR Validation

To confirm the expression of unigenes, 12 unigenes were selected for qRT-PCR. The analysis was performed using samples with tuberous roots at the three stages from two cultivars. The qRT-PCR and data analyses were performed as described by [28]. A total of 12 unigenes, including 4 plant hormone biosynthesis-related genes, 5 lignin biosynthesis-related genes, and 3 transcription factors, from the RNA-Seq were validated, and the primers used for the validation were listed in Appendix A. Sweet potato *Ibactin* gene was used as the reference gene for normalizing quantities of gene expression.

### 2.6. Statistical Analysis

Statistical analysis was performed using SPSS version 13.0 with ANOVA and Duncan’s test for dry matter, lignin, hormone, and qPCR results. Data are means with three biological replicates, with each error bar representing standard error. The statistical significance difference was calculated with Duncan’s new multiple ranges test and marked with asterisks at *p* < 0.05.

## 3. Results

### 3.1. Characteristic of SR Development at Different Stages in Two Cultivars

We analyzed the phenotypes of two varieties (*cv.* Jishu25 (J25) and *cv.* Jishu29 (J29)) at three different time periods (D1, D2, and D3) (Figure 1A), the two varieties had a significant difference in the characteristics of SR. The expanding level, number of expansion SRs, and yield of J29 were significantly higher than J25 at D1, D2, and D3 (Figure 1B–D).

In the D2 stage, the SR of J25 had just begun to expand, but there were already obviously expanded SRs in J29, and the SR number of J29 during the D2 stage was significantly higher than J29. The yield and R/S value also indicate that J29 starts root expanded earlier than J25, and the SR expansion of J29 storage was faster during the D1–D2 stage.

### 3.2. Determination of Lignin Content

Lignin affects SR development in sweet potato. To understate the dynamic changes in lignin accumulation in sweet potato, the lignin content in SR was detected during the root development stages in two cultivars. The results showed that lignin content in SR decreased gradually from D1 to D3 stages in the two cultivars, and the lignin content in J29 was significantly higher than those of J25 at the D1 and D2 stages (Figure 1E).

### 3.3. Global Analysis for RNA-Seq Data

To compare the molecular mechanisms of tuberous root development of both cultivars, the samples were sampled in three stages (D1, D2, and D3). Three independent biological replicates were taken from each stage of each variety, and each biological replicate came from three independent sweet potato SRs (Appendix A). Meanwhile, the RNA of each sample was mixed for SMRT sequencing. Transcriptome sequencing analysis yielded a total of 142.931 GB of clean data, with 7.94 GB of data per sample on average. In addition, Q20 ≥ 96.54% and Q30 ≥ 90.95% were identified in all samples (Appendix A). We obtained 741,084 circular consensus sequencing (CCS) reads ranging from 51 bp to 14,896 bp with an average length of 1322 bp, which included 77.1% of full-length non-chimeric (FLNC) and 20.7% of non-full-length (NFL) reads. After correcting with Hiseq2500 sequencing, 52,137 transcripts and 21,148 unigenes were obtained with an average length of 1253 bp and 1457 bp, which indicates that the data are of high quality (Appendix A). The sequence length after redundancy varied from 53 bp to 6868 bp with the mean length of 1457 bp. Pearson analysis of the transcriptome data found that the three replicates of each line had good consistency and met the requirements of subsequent analysis (excepted for J29_D2_3, which has been deleted) (Appendix A).

### 3.4. Annotation and Classification of Sweetpotato Unigenes

To identify the predictive functions of unigenes in sweet potato SRs, all of the assembled unigenes were matched against the seven databases. Based on the sequence similarity, a total of 96.51% of the unigenes were annotated by alignment in at least one database. Of them, 19,137 (90.49%) unigenes were aligned against the NR databases, 16,083 (76.05%) against SwissProt, 18,849 (89.13%) against KEGG, 11,585 (54.78%) against KOG, 12,949 (61.23%) against GO, 20,114 (95.11%) against NT, and 12,949 (61.23%) against Pfam (Appendix A).

In GO classification, 12,494 unigenes were successfully assigned to 51 functional groups, of which 25 groups belonged to the biological process, 10 to molecular function, and 16 to cellular components. In the category of biological process, the most abundant groups contained metabolic process (6413 unigenes, 51.33%) and the cellular process (6198 unigenes, 49.61%). For molecular function, the highest categories were binding (7133 unigenes, 57.09%) followed by catalytic activity (5943 unigenes, 47.57%). Furthermore, the majority of cellular components were cells (2639 unigenes, 21.12%) and cell parts (2639 unigenes, 21.12%) (Appendix A). For KEGG annotation, 18,849 unigenes were clustered into 44 subcategories. As shown in Appendix A, the groups “the signal transduction pathways” (932 unigenes, 4.94%) and “the translation pathways” (840 unigenes, 4.46%) formed the two largest clusters, followed with the energy metabolism (796 unigenes, 4.22%) and the carbonydrate metabolism (785 unigenes, 4.16%). After KOG annotation, 11,585 unigenes were divided into 24 functional clusters as shown in Appendix A. The top three categories included general function prediction only (1885 unigenes, 16.27%), posttranslational modification, protein turnover, chaperones (1614 unigenes, 13.93%), and signal transduction mechanisms (1185 unigenes, 10.23%).

### 3.5. Analysis of Differential Expression Genes (DEGs)

In this study, we analyzed the global gene expression profiles of sweet potato SR in different developmental stages. According to the false discovery rate and fold change, 14,598 DEGs were screened through cluster analysis in the SRs during the three different developmental stages of two sweet potato cultivars. To examine the gene expression difference during the root development stages in the two genotypes, the DEGs were identified by the comparisons of the nine DEG libraries, i.e., J25-D2 vs. J25-D1 and J25-D3 vs. J25-D2 (Appendix A). The largest number of DEGs occurred between J29-D3 vs. J25-D3 with 3063 up-regulated and 4304 down-regulated unigenes. Furthermore, 6675 and 6571 unigenes were significantly differentially expressed between J25-D3 vs. J25-D2, and J29-D1 vs. J25-D1, respectively. It is interesting that the number of DEGs marked in the two genotypes increased from D1–D2 comparison to D2–D3 (Figure 2). Meanwhile, the number of up- or down-regulated DEGs in Jishu25 was much bigger than those in J29, which revealed that the transcriptome of J25 changed drastically compared to J29 (Figure 2). Moreover, cluster analysis of 14,598 DEGs also showed this result (Figure 3).

### 3.6. Pathway Analysis of DEGs

To determine the involvement of these differentially expressed genes in SRs, KEGG (Kyoto Encyclopedia of Genes and Genomes) pathway enrichment of DEGs was performed in J29-D2 vs. J29-D1 and J25-D2 vs. J25-D1. The upregulated genes in J29-D2 vs. J29-D1 were identified to be involved in 29 distinct metabolic pathways. Of them, the top five were Glycolysis/Gluconeogenesis (ko00010, 9 unigenes), Taurine and hypotaurine metabolism (ko00430, 3 unigenes), Regulation of autophagy (ko04140, 4 unigenes), α-Linolenic acid metabolism (ko00592, 4 unigenes), and Tyrosine metabolism (ko00350, 4 unigenes) (Figure 4A). The upregulated genes in J25-D2 vs. J25-D1 were identified to be involved in 97 distinct metabolic pathways. Of them, the top five were Plant hormone signal transduction (ko04075, 52 unigenes), Regulation of autophagy (ko04140, 10 unigenes), Porphyrin and chlorophyll metabolism (ko00860, 14 unigenes), Circadian rhythm-plant (ko04712, 10 unigenes), Arginine, and proline metabolism (ko00330, 13 unigenes) (Figure 4B).

The down-regulated genes in J29-D2 vs. J29-D1 were identified to be involved in 10 distinct metabolic pathways. Of them, the top five pathways were Protein processing in the endoplasmic reticulum (ko04141, 31 unigenes), Plant–pathogen interaction (ko04626, 4 unigenes), Photosynthesis (ko00195, 3 unigenes), Photosynthesis-antenna proteins (ko00196, 2 unigenes), and Isoquinoline alkaloid biosynthesis (ko00950, 1 unigenes) (Figure 4C). Similarly, the downregulated genes in J25-D2 vs. J25-D1 were identified to be involved in 87 distinct metabolic pathways. The top five pathways were Protein processing in the endoplasmic reticulum (ko04141, 107 unigenes), Endocytosis (ko04144, 43 unigenes), Amino sugar and nucleotide sugar metabolism (ko00520, 30 unigenes), Plant hormone signal transduction (ko04075, 47 unigenes), and Plant–pathogen interaction (ko04626, 31 unigenes) (Figure 4D).

### 3.7. DEGs in Photosynthetic Carbon Fixation

The main energy source of crops is carbon fixation in photosynthesis [29].We analytic comparison J25-D1, J25-D2, J29-D1, and J29-D2, DEGs encoding phosphoenolpyruvate carboxykinase (*IbPEPCK*; transcript9840/f2p0/1895), NAD-dependent malic enzyme (*IbNAD-ME*; transcript4371/f2p0/2292andtranscript5573/f2p0/2208) and ribose-5-phosphate isomerase 3 (*IbRPI3*; transcript27804/f2p0/1131, transcript23852/f2p0/1280 and transcript27066/f14p0/1152) were highly induced in J29-D1 and J29-D2. Furthermore, the genes encoding Glyceraldehyde-3-phosphate dehydrogenase (*IbGADPH*; transcript20086/f4p0/1427, transcript42162/f2p0/565 and transcript10013/f6p0/1899) were found and showed high expression levels in J25-D1 and J25-D2 (Figure 5A, Appendix A).

### 3.8. DEGs in Starch and Sucrose Metabolism

Starch is the main component of dry matter in sweet potato SR, while sucrose is the main form of long-distance transportation of assimilated carbon in sweet potato photosynthesis, and also the main form of sugar accumulation in SR. In this study, 52 DEGs involved in starch and sucrose metabolism were screened during D1 and D2, DEGs were involved in Starch and sucrose metabolism, such as sucrose synthase 3 (*IbSUS3*; transcript28966/f2p0/1073), starch branching enzyme (*IbSBE*; transcript40551/f2p0/612), endoglucanase 3(*IbEG3*; transcript13592/f2p0/1714), β-amylase 2 (IbBAM2; transcript6425/f2p0/2143), and α-Glucan phosphorylases (*Ibα-GPs*; transcript4991/f2p0/2262). These DEGs were expressed at low levels in J25-D1 and J25-D2, while they were highly expressed in J29-D1 and J29-D2. Some DEGs, such as ADP-glucose pyrophosphorylase (*IbAGP*; transcript3208/f2p0/2481), glucose-1-phosphate adenylyl transferase (*IbGlgC*; transcript12385/f2p0/1757), hexokinase-3 (*IbHK3*; transcript4707/f4p0/2291), and fructokinase (*IbFRK6*; transcript16215/f3p0/1586 and *IbFRK7*; transcript12297/f12p0/1740) were highly expressed in J25-D1 and J25-D2 (Figure 5B, Appendix A).

### 3.9. DEGs in Phenylpropanoid Biosynthesis

Lignin is an important secondary metabolite in plants and plays an important biological role in plant growth and development. In two cultivars D1 and D2, a total of 41 DEGs were involved in phenylpropanoid biosynthesis. DEGs of caffeic acid 3-O-methyltransferase (*IbCOMT*; transcript23311/f2p0/1289), caffeoyl-CoA O-methyltransferase (*IbCCoAOMT*; transcript44442/f4p0/451), cinnamoyl-CoA reductase 1 (*IbCCR1*; transcript32620/f2p0/899, transcript15258/f3p0/1636) phenylalanine ammonia-lyase (*IbPAL*; transcript4320/f2p0/2325), and cinnamate 4-hydroxylase (*Ib4CH*; transcript16411/f3p0/1589) all maintained low expression levels in J29D1 and D2. In contrast, peroxidase (*swpd1*; transcript24226/f8p0/1239) anionic peroxidase (*swpa4*; transcript26346/f3p0/1164 and swpa7; transcript24981/f4p0/1221), vinorine synthase (*IbVS*; transcript26274/f4p0/1173), and N-hydroxycinnamoyl/benzoyl transferase (*IbHCBT*; transcript12187/f3p0/1766) were significantly suppressed in J25-D1 and J25-D2 (Figure 5C, Appendix A).

### 3.10. DEGs in Hormone Signal Transduction

Plant hormones are a group of naturally occurring, organic substances which influence physiological processes at low concentrations. Through comparisons between the two stages D1 and D2 of the two cultivars, a total of 122 DEGs were enriched in the hormone signal transduction pathway. DEGs involved in auxin pathway include auxin-responsive protein (*IbIAA1*; transcript35089/f2p0/821, *IbIAA14*; transcript23581/f8p0/1293 and *IbSAUR32*; transcript34493/f5p0/858), jasmonic acid-amido synthetase (*IbJAR1*; transcript6686/f2p0/2094) and auxin transporter-like protein 2 (*IbATL2*; transcript6702/f2p0/2111). *IbJAR1*, *IbATL2,* and *IbIAA* were expressed at low levels in J29D1 and D2, *IbIAA1,* and *IbSAUR32* displayed completely opposite expression patterns in D1 and D2 of the two varietiesin especial, *IbIAA1* was highly expressed in J25D1, while its expression level is low in J29D1; on the contrary, it is highly expressed in J29D2 and low in J25D2. Similar *IbSAUR32* was highly expressed in J29D1, while highly expressed in J25D2.DEGs in cytokinine pathway include histidine kinase 4 (transcript7136/f2p0/2053) and response regulator ARR12-like (transcript2522/f2p0/2595). DEGs in Gibberellin pathway encoding a gibberellin receptor GID1b protein (transcript4234/f2p0/2339). DEGs in Brassinolide pathway encoding BRASSINAZOLE-RESISTANT 1 protein (*IbBRZ1*; transcript20795/f2p0/1413) highly expressed in J29 and low expressed in J25.DEGs in Jasmonic acid pathway-encoding TIFY 10A-likeprotein (transcript33328/f2p0/892), protein PnFL-2 (transcript13073/f2p0/1746 and transcript24853/f8p0/1235), transcription factor MYC2 (*IbMYC2-1*; transcript3100/f2p0/2518 and transcript23305/f2p0/1294, *IbMYC2-2*; transcript3011/f4p0/2510) (Figure 5D, Appendix A).

### 3.11. Differentially Expressed Transcription Factors in Root Development

The development of SRs is regulated by transcription factors (TFs) that control various key gene expressions. A total of 1063 transcription factors were identified in three developmental stages of two varieties. Among them, the largest types were AP2/ERF-ERF (103), bHLH (80), C3H (66), MYB (64), bZIP (61), NAC (56), and WRKY (52) (Figure 6), which were reported to be related to root formation and development [30,31,32]. NAC and WRKY TF are the key regulators of the lignifications of vessel cell differentiation [33,34]. In addition, 47 GRAS and 47 AUX/IAA transcription factors which play a crucial role in gibberellins and auxin signal transduction [35,36].

**Figure 6 genes-14-01263-f006:**
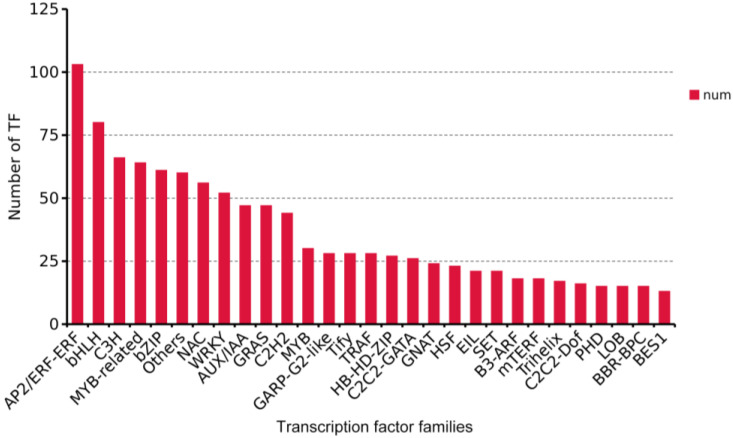
Transcription factors family analysis. The horizontal axis represents the names of different transcription factor families, while the vertical axis represents the number of detected transcription factor families. To identify the DEGs correlating with tuberous root formation and development, WGCNA was implemented to construct a gene network from 27 sweet potato root samples in two cultivars by R package [37]. In the analysis, 17 stable co-expressed modules were obtained through WGCNA (Figure 7A). In the brown module, GO enrichment analysis showed that the unigenes were mainly involved in transferase activity and transcription factor activity in molecular function category, S-adenosylmethionine biosynthetic and metabolic process in biological process category, and apoplast in cellular component (Figure 7B and Appendix A). KEGG enrichment analysis showed that the unigenes were significantly enriched in phenylpropanoid biosynthesis, and plant–pathogen interaction (Figure 7C).

**Figure 7 genes-14-01263-f007:**
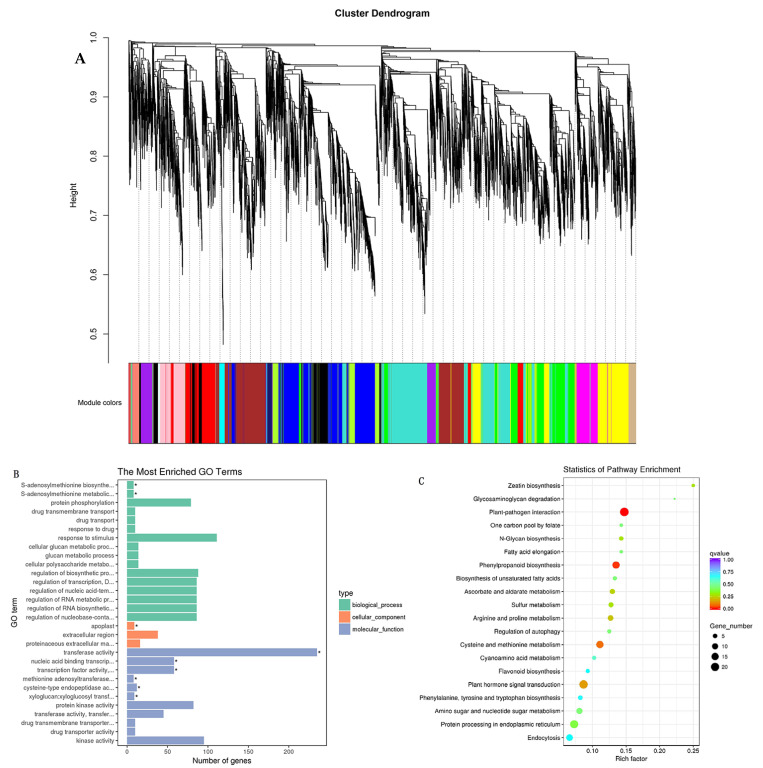
Weighted gene co-expression network analysis (WGCNA) of DEGs. (**A**) Cluster dendrogram of DEGs based on WGCNA analysis. (**B**) GO enrichment analysis of DEGs. (**C**) KEGG enrichment analysis of DEGs.

### 3.12. Validation of RNA-Seq Results

To validate our transcriptome data, qRT-PCR analyses were performed to determine the expression of 12 random DEGs in three root developmental stages of two cultivars. The qRT-PCR results showed that the expression patterns of the 12 DEGs were in good agreement with their RNA-seq results in the root development stages in every cultivar (Figure 8 and Appendix A), and the positive correlation coefficient (R2) was 0.9195. Therefore, the transcriptome data was highly reliable.

## 4. Discussion

SR formation and development are important for sweet potato production. SR expansion is affected by many factors. In this study, we compared two sweet potato cultivar, J25 and J29, and found that J29 started to expand earlier in the early SR period (32–45 days), and its yield was higher (67 days). In order to clarify the differences in SR extension mechanisms between J25 and J29, we carried out comparative transcriptome analysis of two varieties in three stages (D1; 32d, D2; 45d, and D3; 67d), and analyzed the changes of gene expression in three stages. A total of 21,148 genes were identified and annotated into NR, KOG, Pfam, Swiss Prot, and GO databases.

Previous studies have extensively understood the anatomical structure of sweet potato roots. In fibrous roots, the degree of lignification of columnar cells is high, and the activity of vascular cambium is weak, on the contrary, in thick roots, the degree of lignification of columnar cells is high, and the activity of vascular cambium is strong [38].The carbon flux distribution in the starch and lignin metabolic pathways can affect the development of fibrous roots towards pencil roots or SRs [11]. Down-regulation of lignin biosynthesis and up-regulation of starch biosynthesis are the main events leading to the initiation of SRs [9].

In this study, we found that the overall number and yield of SR in J29 were higher than those in J25. In addition, at D2, the root–shoot ratio of J29 was significantly higher than that of J25. The measurement of lignin content showed that After D1 to D3 of transplantation, the lignin content of both varieties decreased, but J29 had a more significant reduction in lignin at D1 and D2 (the lignin content in J29 was significantly higher than that in J25 at D1, and it was slightly lower than that in J25 at D3). Research shows GA promotes lignification and secondary wall formation; SR formation is accompanied by marked reductions in GA signaling [39]. High levels of lignification appear to be detrimental to storage organ formation [40], the application of exogenous GA on sweet potato branches leads to a down-regulation of starch biosynthesis genes, while an up-regulation of lignin biosynthesis genes enhances root lignification, leading to a decrease in SR expanding [41]. Overexpression of the maize LC gene in sweet potatoes stimulates lignin biosynthesis, leading to enhanced lignification of vascular cells in early SRs, severely reducing the expansion of SRs [11]. Therefore, we selected *swbp1* and *swpa7* (peroxidase; transcript24226/f8p0/1239 and transcript26346/f3p0/1164), two DEGs involved in phenylpropanoid biosynthesis, peroxidase is widely involved in plant physiological processes including growth, maturation, seed germination, and regulation and crosstalk of plant hormone signals. It acts downstream of the phenylpropanoid pathway and aggregates lignin monomers to form lignin A [42]; Increased lignin and phenolic content in transgenic plants overexpressing the sweet potato peroxidase gene *swpa4* [43]; Lee found that overexpression of the sweet potato peroxidase gene *IbLfp* increased lignin content in SRs [44]. The quantitative RT-PCR results indicated that *swbp1* and *swpa7* were strongly down-regulated during SR expansion, especially the expression level in J29 was significantly lower than that in J25 (Figure 9A,B and Appendix A). These results indicate that *swbp1* and *swpa7* might play a critical role in SR expansion by reduction of lignin content.

The development of SR and lignin synthesis in plants is regulated by multiple transcription factors. Transcription factors that regulate lignin synthesis have been identified in various plants instantaneously. The AP2/ERF transcription factors (TFs) regulate various processes of plant growth, development, and response to environmental stimuli [45]. Expression of *CsERF1B* in citrus peel can enhance the activity of enzymes related to lignin synthesis, such as POD and COMT, and promote lignin accumulation [46], overexpression of *ERF139* significantly increases the total lignin accumulation in hybrid poplar [47], overexpression of sweet potato ERF transcription factor *IbRAP2.4* inhibits SR formation by activating the expression of genes involved in lignin biosynthesis pathways [48]. We identified two AP2/ERF transcription factors, *IbERF061* and *IbERF109* (transcript22489/f2p0/1329 and transcript28975/f5p0/1069) which down-regulated during SR development, and our qPCR results revealed that the expression of two ERF transcription factors in J29 is significantly higher than those in J25 (Figure 9C,D and Appendix A). This is consistent with the trend of lignin synthesis regulatory genes described above. We speculate that during the SR development of J29, the POD activity in the SRs may be affected by reducing the expression level of *IbERF061* and *IbERF109* transcription factors, resulting in a decrease in lignin content and earlier expansion than J25.

## 5. Conclusions

The rate of SR formation and expansion affects sweet potato yield. In this study, we compared the transcriptomes of SR at the three different development stages in the two cultivars. The results of de novo assembly identified and annotated 52,137 transcripts and 21,148 unigenes were obtained after corrected with Hiseq2500 sequencing. Through the comparative analysis, 9577 unigenes were found to be differently expressed in the different stages of the two cultivars. Phenotypic analysis of two cultivars, combined with analysis of GO, KEGG, and WGCNA showed the regulation of lignin synthesis and related transcription factors plays a crucial role in the early expansion of SR. The four key genes *swbp1*, *swpa7*, *IbERF061,* and *IbERF109* were proved as potential candidates for regulating lignin synthesis and SR expansion in sweet potato. However specific experiments are needed to further verify the function of these genes.

## Figures and Tables

**Figure 1 genes-14-01263-f001:**
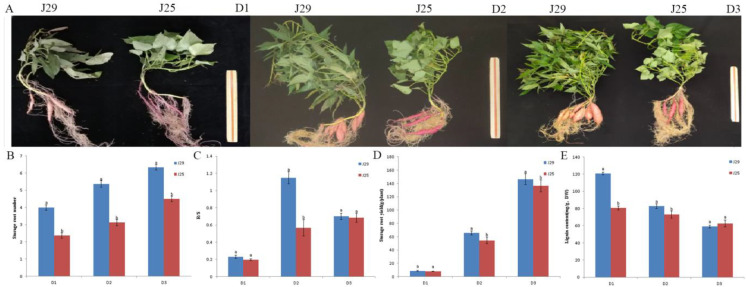
Morphology and characteristics of sweet potato storage roots (SRs) at different stages in two cultivras. (**A**) Phenotypic characterization of SRs at 32 days after planting (DAP) (D1), 46 DAP (D2), and 67 DAP (D3) stages, the length of the ruler is 25 cm. (**B**) SR numbers at different stages. (**C**) The root/shoot (R/S) raito at different stages. (**D**) SR yield at per plant in two cultivras at different stages. (**E**) The lignin content during SR development. Data are means ± SE of three biological repeats, error bars indicate error standard. Means denoted by the same letter were not significantly different at P > 0.05, and different letters indicate statistically significantly differences (*p* < 0.05).

**Figure 2 genes-14-01263-f002:**
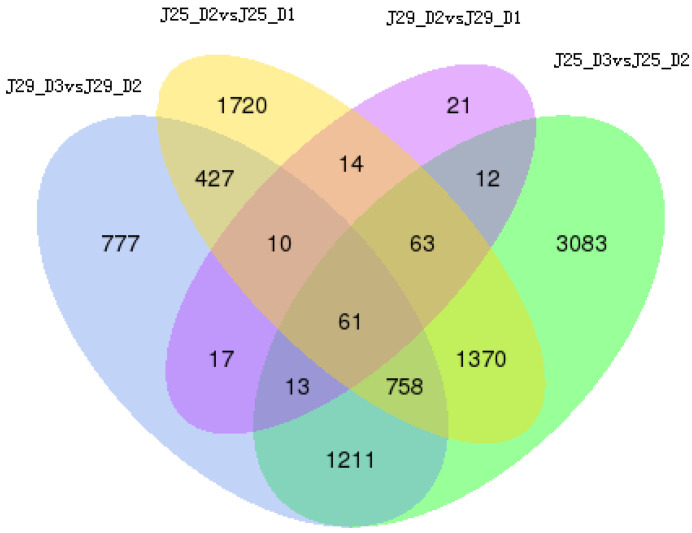
Number of differentially expressed transcripts between D3 and D2 stage (J29-D3 vs. J29-D2), D2 and D1 stage (J29-D2 vs. J29-D1) in J29 SRs, as well as D3 and D2 stage (J25-D3 vs. J25-D2), D2 and D1 stage (J25-D2 vs. J25-D1) in J25 SRs.

**Figure 3 genes-14-01263-f003:**
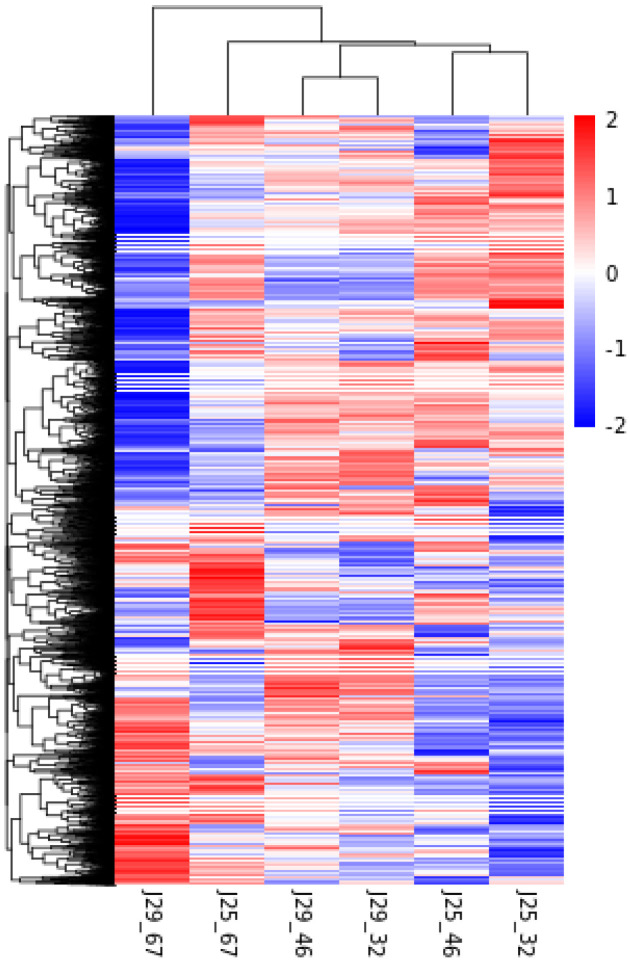
Cluster analysis of differentially expressed genes. The red grids indicate up-regulation of expression, while the blue grids indicate downregulation of expression.

**Figure 4 genes-14-01263-f004:**
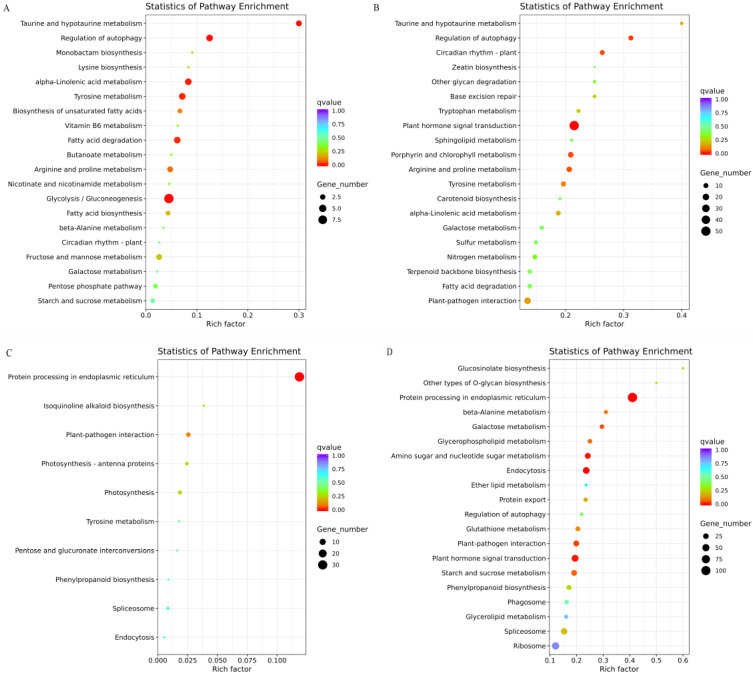
KEGG enrichment of differentially expressed genes (DEGs). The different color depth tables and the size of points represent the Q value and the number of DEGs in that pathway, respectively. (**A**) The up-regulated genes in J29-D2 vs. J29-D1 stage. (**B**) The upregulated genes in J25-D2 vs. J25-D1 stage. (**C**) The downregulated genes in J29-D2 vs. J29-D1 stage. (**D**) The downregulated genes in J29-D2 vs. J29-D1 stage.

**Figure 5 genes-14-01263-f005:**
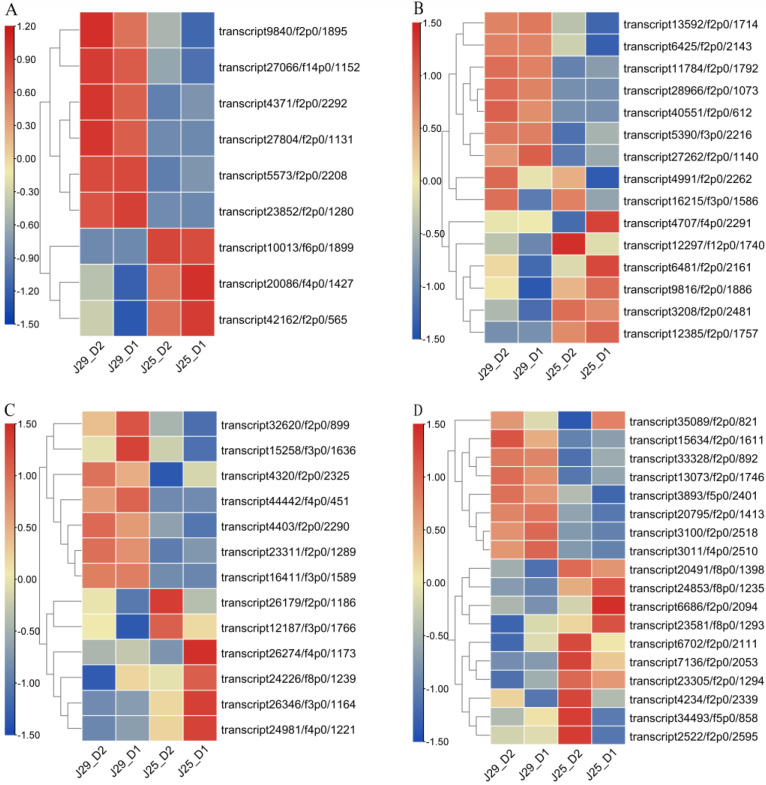
Expression analysis of key DEGs involved in different pathways. Each column represents an experimental condition, and each row represents a gene. Red means the higher expression of a DEG and green means the lower. (**A**) Key DEGs in phosphoenolpyruvate carboxykinase. (**B**) Key DEGs in starch and sucrose metabolism. (**C**) Key DEGs in phenylpropanoid biosynthesis. (**D**) Key DEGs in hormone signal transduction.

**Figure 8 genes-14-01263-f008:**
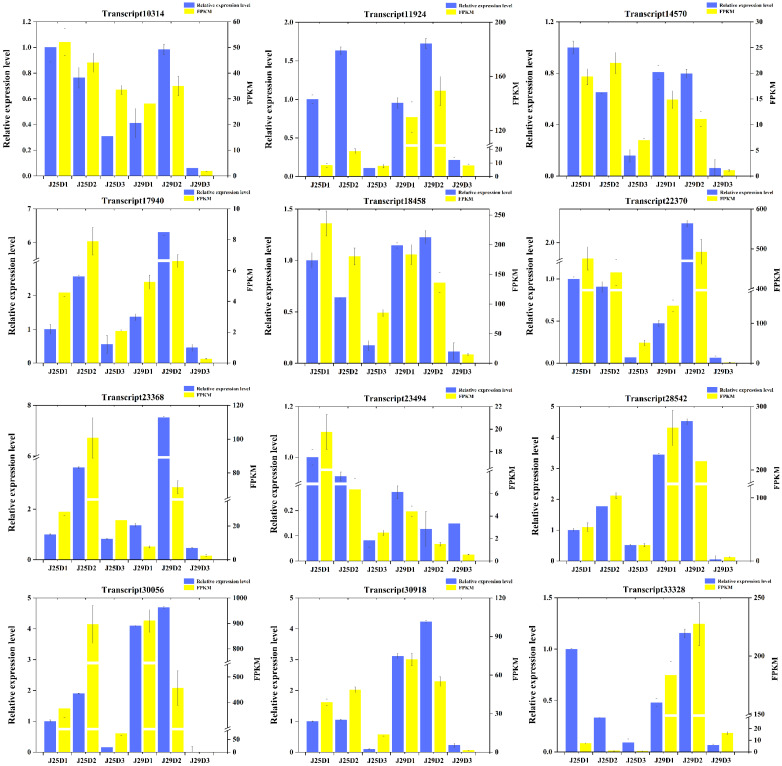
qRT-PCR analysis of differentially expressed gene. The blue columns represent qRT-PCR data, and the yellow columns represent transcriptome data. FPKM values were used to indicate the relative expression of genes in the transcriptome.

**Figure 9 genes-14-01263-f009:**
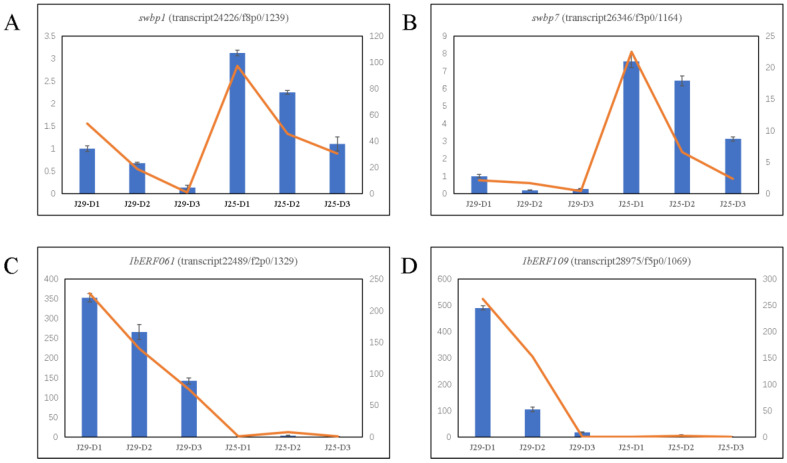
(**A**) The trend of *swbp1* (peroxidase; transcript24226/f8p0/1239) expression in qRT-PCR and transcriptome. (**B**) The trend of *swpa7* (transcript26346/f3p0/1164) expression in qRT-PCR and transcriptome. (**C**) The trend of ERF transcription factor *IbERF061* (transcript22489/f2p0/1329) expression in qRT-PCR and transcriptome. (**D**) The trend of ERF transcription factor *IbERF109* (transcript28975/f5p0/1069) expression in qRT-PCR and transcriptome. Blue bars represent the data of qRT-PCR and orange lines represent the data of transcriptome.

## Data Availability

The trancriptome sequences described here are accessible via GenBank accession numbers PRJNA756699.

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
