# Peer review of "Comparative Transcriptome Analysis Reveals the Effect of Lignin on Storage Roots Formation in Two Sweetpotato (Ipomoea batatas (L.) Lam.) Cultivars"

_genes, 2023, doi:10.3390/genes14061263_

Round 1

Reviewer 1 Report

The manuscript “Comparative Transcriptome Analysis Reveals the Effect of Lignin on Storage Roots Formation in Two Sweetpotato (Ipomoea 3 batatas (L.) Lam.) Cultivars”by Taifeng et al sent for publication to Genes deals with important topic such as lignin synthesis and storage roots formation in sweet potato. The manuscript will be of interest to the scientific community working on the topic. Below is the evaluation report.

Introduction:

The introduction is well written and point the main problems related to the topic.

Materials and methods:

M&M are somehow well written. I just have a question about WGCNA. Did the authors use all the RNA-seq data (all the transcript identified) for the co-expression analysis?

Results:

This part is well written.

Discussion:

This section is well written and supported by results.

Minor remark: Suppl. Table 5 inside the file the authors wrote Suppl. Table 4. This should be corrected. And correct title should be added.

Overall, the manuscript deserved to be published after minor revision.

Author Response

  1. I just have a question about WGCNA. Did the authors use all the RNA-seq data (all the transcript identified) for the co-expression analysis?

Revised: Thanks so much for your comments. In this study we used all the RNA-seq data for the co-expression analysis.

  1. Suppl. Table 5 inside the file the authors wrote Suppl. Table 4. This should be corrected. And correct title should be added.

Revised: Thanks so much for your comments. Suppl. Table 5 has been corrected.

Reviewer 2 Report

In the manuscript “Comparative Transcriptome Analysis Reveals the Effect of Lignin on Storage Roots Formation in Two Sweetpotato (Ipomoea batatas (L.) Lam.) Cultivars" Du et al. performed transcriptomic analysis of storage root of sweet potatoes. The SR samples collected three time points, at 32, 46, and 67 days after planting (DAP), from two cultivars Jishu25 and Jishu29. Then they performed systemic analysis of sequencing results and identified four key genes regulating lignin synthesis and SR expansion in sweetpotato that may affect sweetpotato yield.

The title and abstract are appropriate for the content of the text. The text is easy to follow as the manuscript is well structured with clear logic. The experiments were well conducted.

As explained below, the concerns are listed below.

1.     Line 24 “transcription factors plays a crucial role in the early expansion of SR” the highlighted plays should be play.

2.     Line 32-34, “Sweetpotato (Ipomoea batatas (L.) is one of the seventh important food crops in the world, and produces approximately 88.9 million tons storage root (SR) from the area of 7.4 million ha [1]” does it mean, sweetpotato is the seventh or one of the seven most important food crops?

3.     Fig. 1, please use the same font and size for all panels.

4.     Fig.1 A, please provide the length of bar in each image at the figure legend.

5.     Please provide more information in the figure legends to make it easier to read.

6.     Line 234 the bracelets format is not consistent with the whole draft.

7.     Fig.4, please use bigger sized font.

8.     Fig. 5, please move the A label to the upleft corner of each panel. Also, it is easier for readers to follow if all the transcriptxxxxx/xxxx/xxxx can be replaced by gene name if is it available.

9.     Fig. 6, for the x-axis, it should be the transcription factor families.

10.  Fig.7 B, the names of the enriched GO terms is not complete.

11.  Fig. 8, please refer to the second part of comment 8.

12.  Fig.9, please provide statistical analysis on each panel. 

Minor editing of English language required.
